# Time Course in Ocular Blood Flow and Pulse Waveform in a Case of Ocular Ischemic Syndrome with Intraocular Pressure Fluctuation

**Ryo Yamazaki, Ryuya Hashimoto \*, Hidetaka Masahara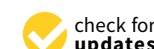, Masashi Sakamoto and Takatoshi Maeno**

Department of Ophthalmology, Toho University Sakura Medical Center, 564-1 Shimoshizu, Sakura 285-8741, Japan; ryou.yamazaki@med.toho-u.ac.jp (R.Y.); hidetaka.masahara@med.toho-u.ac.jp (H.M.); masashi.sakamoto@med.toho-u.ac.jp (M.S.); tmaeno@sakura.med.toho-u.ac.jp (T.M.)
\* Correspondence: ryuuya.hashimoto@med.toho-u.ac.jp; Tel.: +81-43-462-8811; Fax: +81-43-462-8820

**Abstract:** We report on a 70-year-old Japanese man with complaints of worsening left visual acuity who was diagnosed with ocular ischemic syndrome (OIS) associated with internal carotid artery (ICA) stenosis. A gonioscopy examination showed rubeosis iridis and elevated intraocular pressure (IOP) in the left eye (50 mmHg) at the baseline visit. The optic nerve head (ONH) and choroidal blood flow measured by laser speckle flowgraphy (LSFG) was impaired in the left eye compared with that in the right eye. Additionally, the blowout score (BOS), which indicates the variation of the mean blur rate (MBR) during systolic and diastolic periods, was decreased in the left eye. After treatment with an injection of bevacizumab and administration of Rho-associated kinase-inhibitor ripasudil eye drops, both ocular blood flow and BOS in each vascular bed gradually increased along with IOP reduction. The visual acuity also improved. The current case demonstrated increased blood flow and decreased fluctuation of blood flow in the ONH and choroid before and after the treatment in OIS with rubeosis iridis. The LSFG technique is useful to non-invasively assess the ocular circulation and pulse waveform in OIS.

**Keywords:** ocular ischemic syndrome; internal carotid artery stenosis; rubeosis iridis; ocular blood flow; pulse waveform; optic nerve head; choroid; laser speckle flowgraphy

## 1. Introduction

Ocular ischemic syndrome (OIS) is caused by chronic ocular hypoperfusion associated with stenosis of the internal carotid artery (ICA) or the common carotid artery [1]. ICA stenosis strongly affects both retinal and choroidal blood circulation. This is due to the fact that the central retinal artery and the short posterior ciliary artery comes from the ICA [2,3].

OIS is related to poor vision and visual field loss at the time of diagnosis, leading to permanent visual loss [4,5]. Once rubeosis iridis occurs, the already reduced blood flow can be easily disturbed by a higher ocular pressure, which contributes to a worse prognosis [4]. Hayreh et al. reported that carotid artery disease could induce ocular arterial occlusive disease as a result of embolism and hemodynamic disturbance [6]. Although fluorescein and indocyanine green angiography (FA and ICGA) are very useful tests to assess retinal and choroidal circulation in ocular arterial occlusive diseases [2,7], it is not always feasible to perform FA and ICGA due to an adverse reaction [8,9]. Therefore, it is very important to evaluate ocular blood flow in eyes with OIS associated with ICA stenosis using the non-invasive method.

Laser speckle flowgraphy (LSFG-NAVI®, Softcare Co. Ltd., Fukuoka, Japan) is a non-invasive quantitative method; it is very useful for measuring the ONH [10,11], retinal [12,13], and choroidal

blood flow [14,15], based on the changes in the speckle pattern of the laser light reflected from the fundus of the eye (mean blur rate, MBR which is the parameter of blood flow speed) [16]. In addition to MBR, LSFG can analyze the pulse waveforms. The blowout score (BOS) is one of these parameters; it is considered an index of the blood flow that is maintained between heartbeats and is calculated as the difference between the maximum and the minimum MBR as well as the average MBR. High BOS indicates a high constancy of blood flow during the cardiac cycle [16].

The previous study that used LSFG demonstrated that MBR in the ONH and choroid in the affected eye significantly decreased compared with that in the fellow eye in OIS [17]. However, to the best of our knowledge, the time course of the optic nerve head, choroidal blood flow, and pulse waveform using LSFG in OIS with rubeosis iridis is yet to be investigated. Herein, we present a case of OIS associated with ICA stenosis after the treatment in which LSFG was used.

## 2. Case Presentation

A 70-year-old Japanese male presented at the first visit with complaints of worsening visual acuity in his left eye. Two weeks before his first visit, he experienced blurred vision and orbital pain. He visited the department of ophthalmology at a local hospital where it was determined that he had higher intraocular pressure (IOP) and rubeosis iridis in the left eye.

He was referred to Toho University Sakura Medical Center in Sakura, Japan (referred to hereafter as "our hospital") on the next day with the main complaint of worsening visual acuity and orbital pain in the left eye. He had a history of type 2 diabetes, hyperlipidemia, and hypertension without any treatment. At the baseline visit to our hospital, the patient's decimal best-corrected visual acuity (BCVA) in the left and right eyes were 0.4 and 1.0, respectively. The IOP in the left and right eyes were 50mmHg and 16 mmHg, respectively.

The slit-lamp and anterior segment examination of the left eye showed cortical cataract and tortuous neovascularization at the papillary margin of the iris; gonioscopy revealed rubeosis iridis (Figure 1A,B). A fundus examination of the right and left eyes showed mild non-proliferative diabetic retinopathy (NPDR) (Figure 1C,D). An optical coherence tomography (OCT) (Spectralis OCT; Heidelberg Engineering Inc., Heidelberg, Germany) examination showed no abnormal findings such as macular edema or nerve fiber layer defect in the ONH and the macular region (Figure 1E–H).

We measured MBR and BOS in the ONH tissue and vessel, and the choroid of the macular region using LSFG. We set the measurement circle at the center of the ONH and the macula in both the right and left eyes at the baseline visit while comparing the fundus photographs and the LSFG color composite map images (Figure 2A–D). Baseline MBRs in the ONH tissue, the ONH vessel, and the choroid in the left and right eyes were 7.5/13.4, 15.0/58.1, and 6.7/15.6, respectively. Baseline BOSs in the entire ONH and in the choroid in the left and right eyes was 47.1/65.4 and 47.2/ 65.6, respectively.

Subsequently, we performed FA and ICGA to examine the time of the choroidal flush, the arm-to-retina, and the retinal circulation. Although the FA findings showed no retinal non-perfusion areas, the time of the choroidal flush was 20 s (Figure 2E), the arm-to-retina was 26 s (Figure 2F), and the retinal circulation was 22 s, suggesting a delay. Findings from the FA examination of the anterior segment in the left eye showed massive fluorescein leakage from the rubeosis iridis (Figure 2G). ICG findings also revealed a delay in the arm-to-choroid time of 19 s after the injection (Figure 2H).

In order to evaluate the ICA stenosis, a carotid ultrasound was performed, and it showed 76% stenosis of the left ICA. The laboratory blood test findings showed that the hemoglobin A1c (HbA1c) was 9.5% and the total cholesterol level was 285 mg/dl, suggesting that they were elevated. We referred the patient to the department of neurosurgery and internal medicine, where it was determined that ICA stenting was not necessary for the stenosis.

The patient was diagnosed with OIS associated with the left ICA stenosis based on the above mentioned clinical findings and thus treatment was initiated. The regimen entailed one injection of bevacizumab (IVB), administration of Rho-associated kinase inhibitor eye drops (0.4% ripasudil),

acetazolamide (750 mg per day) for three days, metformin tablets (500 mg per day), and combination tablets of low-dose aspirin (100 mg) with lansoprazole (15 mg).

The time course of MBR and BOS in each vascular bed in the affected eye is shown in Table 1.

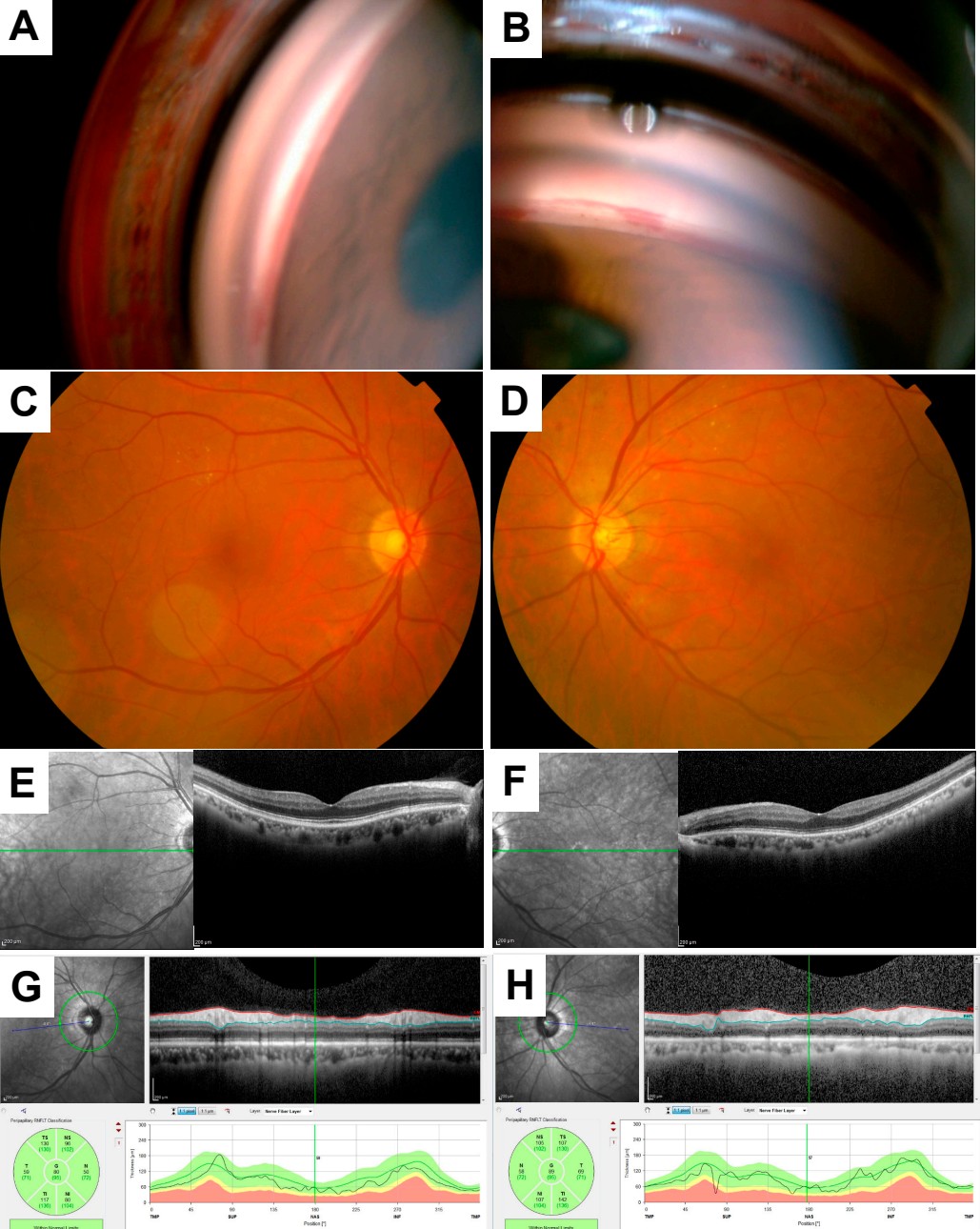

**Figure 1.** Gonioscopy, color fundus photography, and optical coherence tomography findings at the baseline visit. A gonioscopy photo of the left eye (**A**) and rubeosis iridis (**B**). Color fundus photography showed that the right (**C**) and left (**D**) eyes were normal. Optical coherence tomography (OCT) findings of the right (**E**) and the left (**F**) eyes showed no abnormal findings in the macular region. OCT findings of the right (**G**) and the left (**H**) eyes also showed normal finding.

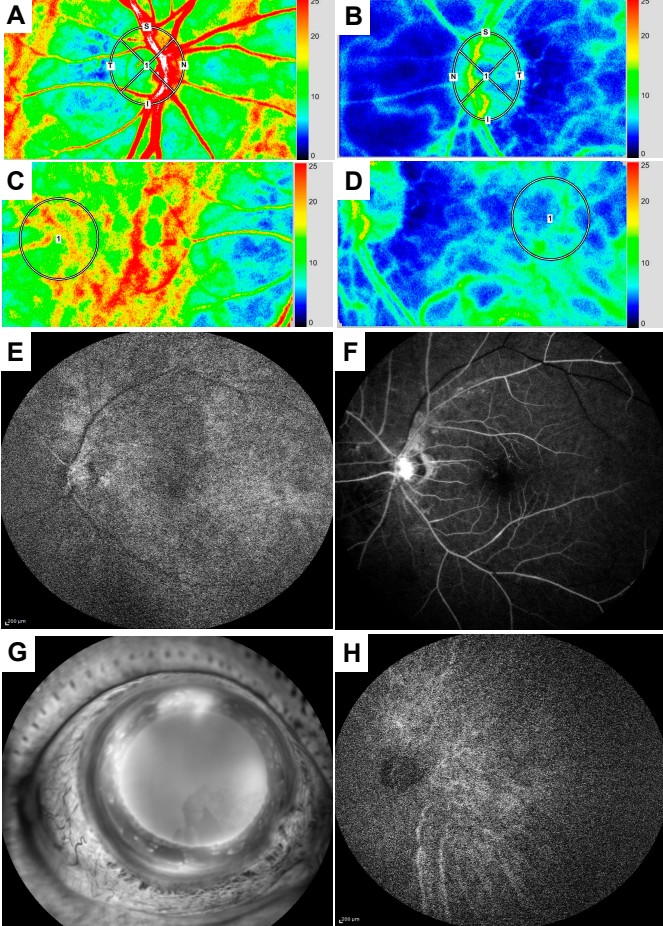

**Figure 2.** Laser speckle flowgraphy, fluorescein, and indocyanine green angiography findings from the baseline visit. Laser speckle flowgraphy (LSFG) findings in the optic nerve head (ONH) of the right (**A**) and left (**B**) eyes. LSFG findings in the macular of the right (**C**) and left (**D**) eyes. Fluorescein angiography (FA) findings in the left eye showed the prolonged times for the following: the choroidal flush was 20 s (**E**), the arm-to-retina was 26 s (**F**), and retinal circulation was 22 s. FA findings of the anterior segment in the left eye showed massive fluorescein leakage from the rubeosis iridis (**G**). ICGA findings also revealed the delay of the arm-to-choroid circulation time which was 19 s (**H**).

Figure 3 shows the time course for the LSFG color composite map of the affected eye.

**Table 1.** Time course of mean blur rate and blowout score in the optic nerve head and choroid in the left eye. MBR; mean blur rate, AU; arbitrary units, ONH; optic nerve head, BOS; blowout score.

|  | MBR in the ONH Vessel (AU) | MBR in the ONH Tissue (AU) | MBR in the Choroid (AU) | BOS in the Entire ONH (AU) | BOS in the Choroid (AU) |
|---|---|---|---|---|---|
| Baseline | 15.0 | 7.5 | 6.7 | 47.1 | 47.2 |
| 4 days | 15.4 | 7.7 | 5.1 | 58.2 | 49.0 |
| 2 weeks | 19.5 | 8.4 | 8.7 | 69.2 | 61.0 |
| 3 months | 20.4 | 8.5 | 9.0 | 74.3 | 65.4 |
| 5 months | 22.2 | 8.4 | 8.7 | 67.0 | 64.2 |

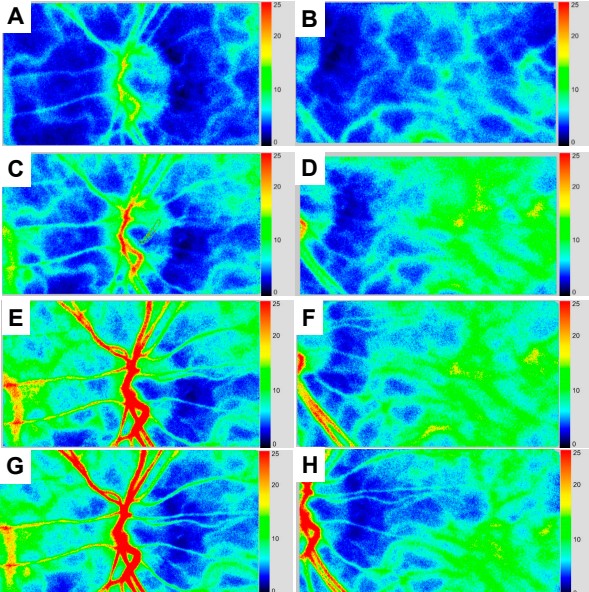

**Figure 3.** The time course exhibited by the laser speckle flowgraphy color composite map. LSFG images in the optic nerve head and the choroid of the left eye at 4 days (**A**,**B**), 2 weeks (**C**,**D**), 3 months (**E**,**F**), and 5 months (**G**,**H**) after the baseline visit. The LSFG images showed that warm colors increased over time. The Mean blur rate and blowout scores in each vascular bed also increased after treatment.

Four days after the baseline visit, anterior segment examination and gonioscopy showed reduced rubeosis iridis, and the IOP in the affected eye decreased to 41 mmHg. However, the LSFG findings showed no improvement in blood flow in each ocular vascular bed when compared to the baseline visit (Figure 3A,B).

Two weeks after the baseline visit, the IOP in the affected eye reduced to 15 mmHg, and the decimal BCVA in the left eye improved to 0.6 without visual field defect based on the Goldman visual field test. Gonioscopy showed the near disappearance of the rubeosis iridis. LSFG findings in the left eye indicated that warm colors were increased compared to the findings from the baseline visit (Figure 3C,D).

Three months after the baseline visit, the left decimal BCVA and IOP were 0.6 and 21mmHg using 0.4% ripasudil, respectively. The anterior segment examination of the left eye and gonioscopy showed no rubeosis iridis (Figure 4A,B), and the LSFG results demonstrated that warm colors were more prevalent when compared to that from the baseline visit. MBR in the ONH tissue and vessel increased from 15.0 to 20.4 and 7.5 to 8.5, respectively. Choroidal blood flow also increased from 6.7 to 9.0 (Table 1, Figure 3E,F). In addition to MBR, BOS in the entire ONH and in the choroid also significantly increased from 47.1 to 74.3 and 47.2 to 65.4, respectively. Figure 5 shows the changes in pulse waveform for the entire ONH and choroid between the 3 month and baseline visit; it showed less fluctuation in the ONH and choroidal blood flow 3 months after treatment initiation. We thus repeated the FA and ICGA; the FA findings for the left eye showed no delays. The choroidal flush occurred in 16 s (Figure 4C), the arm-to-retina time was 18 s (Figure 4D), and the retinal circulation time was 10 s. There was no leakage in the left anterior segment (Figure 4E). The ICGA findings at 19 s after the injection (Figure 4F) show improvements in choroidal circulation when compared to that from the baseline visit.

Five months after the baseline visit (the last visit), the IOP in the affected eye was 18 mmHg and the BCVA in the affected eye was 0.6 without any ophthalmic treatment. Laboratory blood tests showed reductions in the HbA1c (7.3%) and the cholesterol (181mmg/dl) levels. LSFG findings demonstrated sufficient blood flow and higher BOS in each vascular bed compared with that at the last visit (Table 1,

Figure 3G,H). During the follow-up periods, the patients did not receive the retinal photocoagulation and revascularization surgery.

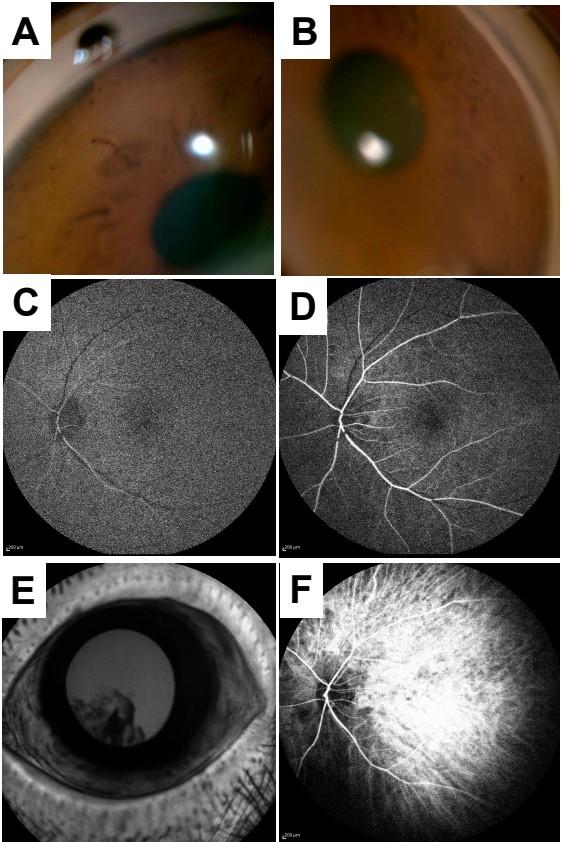

**Figure 4.** Gonioscopy, fluorescein, and indocyanine green angiography findings 3 months after the baseline visit. A gonioscopy photo of the left eye (**A**) and showing no rubeosis iridis (**B**). Fluorescein angiography (FA) of the left eye shows no delays in the following: the choroidal flush was 16 s (**C**), the arm-to-retina time was 18 s (**D**), and the retinal circulation time was 10 s. There was no leakage in the left anterior segment (**E**). An indocyanine green angiography (ICGA) at the early phase was 19 s (**F**).

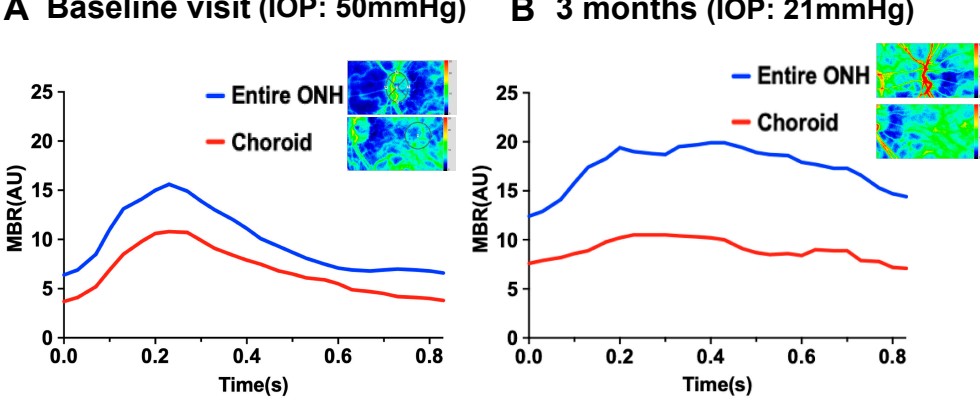

**Figure 5.** Changes in the pulse waveform in the entire optic nerve head and the choroid at baseline and 3 months after treatment. Pulse waveform in the entire optic nerve head (ONH) and the choroid at baseline. Intraocular pressure (IOP) was 50mmHg (**A**). Pulse waveform in the entire ONH and the choroid 3 months after treatment. IOP was 21mmHg (**B**). Blue color indicates the entire ONH, and red color indicates the choroid.

In the current case, we measured the IOP and patient's systolic and diastolic blood pressure (SBP and DBP) for calculation of the mean blood pressure (MBP) and OPP at every visit [18]. OPPs at the baseline, 2 weeks, 3 month, and 5 month visits in the left and right eyes were 31.3 and 65.3 mmHg, 65.7 and 64.7 mmHg, 61.6 and 67.6 mmHg and 60.8 and 61.8 mmHg, respectively.

Subject gave informed consent for inclusion before they participated in the study. The study was conducted in accordance with the Declaration of Helsinki, and the protocol was approved by the Ethics Committee of Toho University Sakura Medical Center (number: 2015056).

## 3. Discussion

In the present case, we revealed the time course of the blood flow and pulse waveform in the ONH and choroid along with IOP reduction using LSFG before and after the treatment in OIS. To our knowledge, this is the first time this has been completed.

At the baseline visit, the ocular blood flow in the ONH tissue and vessel area, and the choroid of the left eye was decreased compared with that in the fellow eye. Additionally, the degree of ICA stenosis was 76 %. This finding is similar to that from a previous study in which the degree of the ICA stenosis and the ratio of the affected to the fellow eye was shown to be inversely correlated [17]. In the present case we found a higher level of HbA1c and IOP (50mmHg) caused by rubeosis iridis. It was previously demonstrated that type 2 diabetes [19], hypertension, and hyperlipidemia [20] are associated with impaired ONH blood flow autoregulation in both vascular and tissue areas in response to elevated IOP. Additionally, the degree of autoregulation in the ONH was negatively correlated with the HbA1c level. Akahori et al. reported that there was a significant correlation between the percent reduction for the OPP and the choroidal MBR during the elevation of IOP by 30 mmHg, even among healthy subjects [21]. Based on the above findings, it is understandable that the decreased blood flow in each vascular bed is mainly associated with impaired ONH and choroidal blood flow autoregulation due to the decreased OPP in addition to the ICA stenosis.

The current case showed that MBR in each vascular bed gradually increased along with reduced IOP compared to that at baseline. After treatment for rubeosis iridis using the IVB injection and 0.4% ripasudil ophthalmic solution, OPP was significantly increased twofold by a reduction in IOP. Riva et al. reported that there is a linear relationship between choroidal blood flow and OPP in healthy subjects [22]. Another study also demonstrated that the percent change in the OPP and MBR showed positive linear correlations in the choroid and the ONH [23]. In this case, the patient was given an antiplatelet tablet and ripasudil eye drops. The antiplatelet is known to increase pulsatile ocular blood flow among those with diabetic retinopathy [24]; ripasudil has also been reported to increase ONH blood flow in the eyes of normal rats [25,26]. The level of the HbA1C and cholesterol reduced several months after the initial visit. The pulsatile ocular blood flow is influenced by changes in plasma glucose concentrations in Type II diabetes mellitus [27]. In light of these findings and that incremental OPP due to decrement IOP, the improvement of the general condition and these administrative treatments might improve ONH and choroidal blood flow.

In addition to MBR, the present case showed lower BOS, which means a greater fluctuation in blood flow, in each vascular bed at the baseline visit. The BOS gradually increased after the treatment over the follow-up period. The graph of the pulse waveforms in the entire ONH and the choroid (Figure 4) showed less fluctuation in blood flow 3 months after treatment, compared with that at the baseline visit.

A previous study found that BOS was decreased among those with a high IOP [23]. In the current case, ripasudil was reported to suppress the activity of rho/rock kinase two and further increased the endothelial nitric oxide synthase activity [28], which is crucial for nitric oxide (NO)-mediated vasodilation of vessels after arterial occlusion [29]. Additionally, metformin improved the endothelial function and reduces blood pressure via the up-regulation of NO in the diabetic animal model [30]. Considering the above and the fact that BOS is an index of the blood flow that is maintained between heartbeats, the lower BOS of the eye with the OIS suggests that ocular circulation was unstable

and insufficient when the IOP was elevated. Besides, the increase in BOS might be related to the improvements in vasodilation due to these administrative treatments.

Lastly, in the current case there were increases in MBR and BOS in each vascular bed as visualized using LSFG, and there was a shortened circulation time for the choroidal and the retina obtained using FA and ICGA. Nagasato et al. reported that MBR was significantly correlated with the arteriovenous passage time obtained by FA [31]. From the above findings, LSFG might have the potential to assess these circulation times instead of FA and ICG. However, further studies are needed to evaluate whether the changes in these circulation times using FA and ICGA are associated with those in the MBR and those obtained by the LSFG displayed using the pulse wave.

In conclusion, the current case demonstrated increased blood flow and decreased fluctuation of blood flow in the ONH and choroid along with IOP reduction. The LSFG technique is useful to non-invasively assess the time course in ocular circulation and pulse waveform in OIS.

**Author Contributions:** Conceptualization: R.H.; investigation: R.Y., R.H., H.M., and M.S.; writing—original draft preparation: R.Y. and R.H.; writing-review and editing: R.H. and T.M. All authors have read and agreed to the publication of the final version of this manuscript.

**Funding:** This research received no external funding.

**Conflicts of Interest:** The authors declare no conflicts of interest.

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
