# Peer review of "Time Course in Ocular Blood Flow and Pulse Waveform in a Case of Ocular Ischemic Syndrome with Intraocular Pressure Fluctuation"

_2411-5150, 2020_

Round 1

Reviewer 1 Report

General comments

Yamazaki et al. described time course in blood flow in a case of ocular ischemic syndrome.  After successful treatment, IOP decreased, and blood flow improved.  Their data are important & should have impact on most ophthalmologists.

I have following comments,

Specific comments

Although the improvement in ocular blood flow seems mostly due to decrease in IOP, improvement in general condition may have some role.  Please discuss. 

Author Response

Reviewer: 1

General comments

Yamazaki et al. described time course in blood flow in a case of ocular ischemic syndrome. After successful treatment, IOP decreased, and blood flow improved. Their data are important & should have impact on most ophthalmologists.

Response: We appreciate the kind comments on the value of our manuscript. We have responded to the comment and have revised the manuscript to address the reviewer’s concerns.

Please find below response to the comment.

Specific comments

  1. Although the improvement in ocular blood flow seems mostly due to decrease in IOP, improvement in general condition may have some role. Please discuss.

Response: According to the reviewer’s suggested, the general condition might contribute to the improvement in ocular blood flow. We have added this point in the 3rd paragraph of discussion (Page 9, lines 214-216) and added the new reference about that (new Refs.27). 

Reviewer 2 Report

The authors evaluated changes in blood flow and pulse waveform of the blood flow in the fundus using laser speckle flowgraphy in a patient with ocular hypertension associated with ocular ischemic syndrome (OIS). Blood flow velocity and fluctuation of the blood flow at the optic disc and the macula increased and decreased, respectively together with decreased intraocular pressure following anti-hypertensive treatment. These results suggest that LSFG is a non-invasive tool to evaluate ocular circulation of this disease with ocular hypertension.

This is an interesting manuscript evaluating detailed functions of choroidal circulation hemodynamics in patients with ocular hypertension. I have a few questions.

Major points

  • The day measured LSFG first (baseline) appears to be the day at the first visit. At the first visit, the present case had high intraocular pressure (IOP) in the affected eye. When IOP is generally very high, MBR values in the ocular tissues would be lower than values at normal IOP because of decreased ocular perfusion pressure. In the present case, the initial MBR reduction (at the first visit and 4 days later) in the affected eye may be mainly due to high IOP. On the other hand, MBR in the entire fundus is likely to decrease in the acute stage of OIS patients without ocular hypertension (reference 17). In the present case, do the authors think that changes in MBR and BOS in the acute phase were affected by OIS itself?

Minor points

  1. Page 2, lines 7-6 from the bottom. “tortuous neovascularization at the papillary margin”; this is findings in the iris?
  2. Please describe whether this case received retinal photocoagulation or revascularization surgery.

Author Response

Reviewer: 2

The authors evaluated changes in blood flow and pulse waveform of the blood flow in the fundus using laser speckle flowgraphy in a patient with ocular hypertension associated with ocular ischemic syndrome (OIS). Blood flow velocity and fluctuation of the blood flow at the optic disc and the macula increased and decreased, respectively together with decreased intraocular pressure following anti-hypertensive treatment. These results suggest that LSFG is a non-invasive tool to evaluate ocular circulation of this disease with ocular hypertension.

This is an interesting manuscript evaluating detailed functions of choroidal circulation hemodynamics in patients with ocular hypertension. I have a few questions.

Response: We greatly appreciate the kind comments on the value of our manuscript. We have responded to all comments and have revised the manuscript to address the reviewer’s concerns. Below please find our point-by-point responses to the comments.

Major points

The day measured LSFG first (baseline) appears to be the day at the first visit. At the first visit, the present case had high intraocular pressure (IOP) in the affected eye. When IOP is generally very high, MBR values in the ocular tissues would be lower than values at normal IOP because of decreased ocular perfusion pressure. In the present case, the initial MBR reduction (at the first visit and 4 days later) in the affected eye may be mainly due to high IOP. On the other hand, MBR in the entire fundus is likely to decrease in the acute stage of OIS patients without ocular hypertension (reference 17). In the present case, do the authors think that changes in MBR and BOS in the acute phase were affected by OIS itself?

Response: We appreciate for the reviewer’s constructive comment. The previous study (reference 17) showed that there is a negative correlation between the degree of ICA stenosis and optic nerve head tissue blood flow. Focusing on the individual cases in the previous study, in the case with the 84 % stenosis of ICA, there was no difference in blood flow at the margin of disc or entire area between the affected and fellow eye in the previous study. In the present case, the degree of ICA stenosis was 76 percent at the initial visit, and MT in the ONH in the affected eye (7.5) was almost half of that in the fellow eye (13.4). In light of these findings, we think that MBR and BOS in the acute phase were mainly affected by higher IOP, although OIS itself might influence these parameters. We added this point to 2nd paragraph of the discussion section.

Minor points

1.Page 2, lines 7-6 from the bottom. “tortuous neovascularization at the papillary margin”; this is findings in the iris?

Response: As reviewer’s comment, tortuous neovascularization existed in the papillary margin of the iris. We have added this point to the 3rd paragraph of the case presentation (page 2, lines 73).

2.Please describe whether this case received retinal photocoagulation or revascularization surgery.

Response: During the follow-up periods, non-perfused area did not appear, and the patients did not receive the retinal photocoagulation and revascularization surgery. Following the Reviewer’s instructive comment, we have added this point to the section of the case presentation (page8, lines 180-181).
